# A Rare Co-Occurrence of Maffucci Syndrome and Astrocytoma with IDH1 R132H Mutation: A Case Report

**DOI:** 10.3390/medicina59061056

**Published:** 2023-05-31

**Authors:** Nurali Ashirov, Iroda Mammadinova, Aidos Moldabekov, Berik Zhetpisbaev, Daniyar Teltayev, Nurzhan Ryskeldiyev, Serik Akshulakov

**Affiliations:** 1Minimal Invasive Neurosurgery Department, National Centre for Neurosurgery, Astana 010000, Kazakhstan; 2Brain Neurosurgery Department, National Centre for Neurosurgery, Astana 010000, Kazakhstan; 3Department of Pathology, National Centre for Neurosurgery, Astana 010000, Kazakhstan

**Keywords:** astrocytoma, IDH-mutation, multiple enchondroma, Maffucci syndrome

## Abstract

*Background*: Maffucci syndrome is a rare genetic disorder associated with the development of multiple enchondromas and soft tissue cavernous hemangiomas, as well as an increased risk of malignant tumors. *Case Description*: Here we report a case of Maffucci syndrome in a patient who presented with a giant left frontal lobe tumor. Molecular genetic analysis of the tumor revealed an isocitrate dehydrogenase (IDH) mutation p.R132H (c.395C>A) mutation in the IDH1 gene and a heterozygous duplication of the CDKN2A genes. *Conclusions*: The presence of an IDH1 mutation is notable because this mutation is frequently seen in glial tumors and other neoplasms, and its co-occurrence with Maffucci syndrome may represent a novel risk factor for the development of gliomas. This case underscores the importance of genetic testing in patients with Maffucci syndrome who present with central nervous system tumors, as well as the need for further research to understand the relationship between IDH1 mutations and the development of gliomas in this population.

## 1. Introduction

Enchondromatosis is a rare skeletal disorder that is characterized by the presence of multiple benign enchondromas that affect the metaphyses of the bones. These enchondromas can cause considerable deformities in the affected area, as well as multiple asymmetric edemas with unilateral prevalence. Ollier disease and Maffucci syndrome are the two most common subtypes of enchondromatosis, both of which are typically diagnosed in the first decade of life [1,2,3]. Apart from the presence of enchondromas, Maffucci syndrome is characterized by several vascular anomalies, such as hemangiomas of the skin and soft tissues, particularly in the limbs and abdominal wall, as well as lymphangiomas [3,4]. Enchondromas and hemangiomas in Maffucci syndrome have a higher likelihood of transforming into malignant tumors and tend to be diagnosed in adults who are over the age of 30 [2,4,5]. The exact cause of enchondromas remains unknown, but recent studies have shown that somatic mutations in metabolic enzymes, specifically isocitrate dehydrogenase 1 and 2 (IDH1/2), are a common observation in the development of enchondromas [2,6,7]. 

Significant advancements in cancer genetics over the past decade have revealed that genes encoding IDHs are frequently altered in various types of human malignancies, especially gliomas. Numerous studies have demonstrated that IDH mutations play a significant role in altering cellular physiology, resulting in modifications to cellular metabolism, changes to epigenomes, and abnormal regulation of redox homeostasis [7,8]. 

The co-occurrence of Maffucci syndrome with glial tumors, specifically astrocytomas, is an extremely rare phenomenon. The association between the two conditions is not fully understood, but some studies have suggested that the IDH mutations present in Maffucci syndrome may predispose individuals to the development of certain types of tumors, including astrocytomas. 

Corvino et al. reported that the medical literature contains over thirty documented cases of glioma associated with Ollier disease, whereas the number of reported cases of astrocytoma in combination with Maffucci syndrome is significantly fewer, and the majority of those cases were described before the era of molecular genetics [9]. 

The available evidence suggests that gliomas in patients with enchondromatosis share similar characteristics with cartilaginous tumors and are likely caused by somatic IDH mosaicism. The timing of IDH mutation acquisition may also impact the molecular features and location of the gliomas. Patients with enchondromatosis-related gliomas tend to develop the tumors at a younger age and have a higher frequency of multicentric tumors. The molecular profile of these gliomas reveals IDH mutations and loss of ATRX expression, while no co-deletion of 1p/19q is observed, which is different from sporadic IDH-mutated gliomas [5].

The rarity of this co-occurrence poses a significant challenge for diagnosis and treatment. Due to the limited number of reported cases, there is a lack of standardized protocols for managing these patients, and the optimal treatment approach remains unclear. However, the identification of the common IDH mutations in both conditions presents a potential avenue for targeted therapy. As genetic studies on the co-occurrence of astrocytoma and Maffucci syndrome are rare, we present a confirmed case of astrocytoma associated with Maffucci syndrome through molecular genetic analysis.

## 2. Case Presentation

A 32-year-old female, previously diagnosed with Maffucci syndrome, presented to the hospital with a three-month history of headaches and weakness in the right upper and lower extremities. There was no birth record or family history of congenital defects, brain tumors, or bone disease. The patient had several enchondromas on her left humerus and forearm, both lower extremities. The patient had undergone ten tumor resection surgeries for multiple hemangiomas located on both sides of her abdominal wall, hand, and foot and also suffered from lymphedema affecting her right hand and foot (Figure 1). Eight years ago, she was diagnosed with a left-sided ovarian cyst and underwent a left oophorectomy, and two months before admission, she underwent pregnancy termination due to multiple fetal malformations. 

Physical examination revealed multiple enchondromas affecting her left humerus, forearm, and both lower extremities, an S-shaped spinal deformity, an O-shaped deformity and lower extremities hypoplasia, as well as left upper limb hypoplasia. The patient presented with right-sided hemiparesis, and a subsequent neurological examination revealed the same. 

A magnetic resonance imaging (MRI) scan showed a massive (83 × 58 × 62 mm) lesion in the left frontal lobe, invading the corpus callosum, with heterogeneous hyperintense T2 and FLAIR signals, hypointensity on T1-weighted images, and peritumoral edema. After gadolinium administration, the lesion showed heterogeneous enhancement on T1-weighted imaging. The preoperative radiological diagnosis was high-grade glioma (Figure 2).

The CT scan of the chest revealed a large, well-circumscribed, destructive lytic lesion (11.7 × 13.1 × 11.2 mm) that had extended into the left lung and mediastinum, confirmed as chondroma by histological analysis. Additionally, a CT scan demonstrates numerous lesions on the sternum, left 5, 6, and 7 ribs, scapula, and humerus (on both sides) with a maximal size of 60 × 49 × 44 mm. There are numerous nodular lesions with calcifications on the anterior chest wall, the largest of which can measure up to 2.3 × 1.0 cm (Figure 3).

The patient underwent frontoparietal craniotomy for subtotal removal of the tumor. Follow-up brain MRI conducted on the second day after the surgery revealed a considerable reduction in tumor size, and neurological examinations showed improvement in the right-sided hemiparesis. 

A pathological examination was carried out using an Axioskop 40 microscope by Carl Zeiss (Oberkochen, Germany) and a Pannoramic MIDI scanning microscope, with a total magnification of ×40, ×100, and ×200. The pathological examination with hematoxylin- and eosin-stained slides showed that the tissue was hypercellular, with nuclei that varied in shape and size and had hyperchromatic and polymorphic features (Figure 4). Additionally, the presence of microcysts, glomerular proliferations of vessels, and foci of coagulation necrosis was noted. 

The molecular genetic study identified a p.R132H (c.395C>A) mutation in the IDH1 gene and a heterozygous duplication of the CDKN2A genes, which confirmed the diagnosis of astrocytoma, IDH-mutant, WHO grade 4, ICD-O code 9445/3.

The neurological deficit regressed during the postoperative period, and the patient’s symptoms gradually improved. On the tenth day following surgery, the patient was discharged for postoperative radiation and chemotherapy.

## 3. Discussion

This case represents a rare manifestation of Maffucci syndrome, where multiple enchondromas can be associated with the development of glial tumors in the central nervous system. 

Enchondromas are benign cartilaginous tumors that arise from the medulla of bones, typically in the metaphysis. Enchondromas are the most common type of benign bone tumor and usually do not cause any symptoms.

However, when multiple enchondromas occur, they can be a sign of a more serious condition, such as Ollier disease or Maffucci syndrome. Ollier disease is a rare disorder characterized by the development of multiple enchondromas in different bones, typically unilateral in distribution with a predilection for the appendicular skeleton. The onset of Ollier disease usually occurs in the first decade of life, but it can occur in early adolescence or adulthood.

Maffucci syndrome is a rare disease identified by the development of multiple enchondromas with vascular anomalies. In addition to multiple enchondromas, individuals with Maffucci syndrome may also develop soft tissue hemangiomas, which are abnormal growths of blood vessels. The hemangiomas can be present on the skin or internal organs [4].

Both Ollier disease and Maffucci syndrome can cause significant symptoms and complications. Multiple swellings on the extremities can lead to deformity around the joints, limitations in joint mobility, scoliosis, bone shortening, leg-length discrepancy, gait disturbances, pain, loss of function, and pathological fractures. Treatment of these conditions is typically aimed at managing the symptoms and preventing complications. 

The co-occurrence of Maffucci syndrome and a glial tumor is an uncommon phenomenon. In individuals with Maffucci syndrome, the occurrence of a glial tumor may present unique challenges in diagnosis and treatment due to the presence of multiple enchondromas and hemangiomas throughout the body.

It has been found that both gliomas and chondromas share a common IDH1/2 mutation mechanism. This discovery has led to a molecular analysis of brain tumor cases associated with multiple enchondromas. About 80% of individuals diagnosed with Ollier disease and Maffucci syndrome have somatic mutations in IDH1 and IDH2 detected exclusively in their tumors, which include enchondromas, chondrosarcomas, and vascular anomalies [10]. 

Skull base enchondromas, chondrosarcoma, and non-mesenchymal neoplasms such as glioma (less than 1/100,000 cases) represent the majority of intracranial involvement [10,11,12,13]. Additionally, a rare case of pituitary adenoma associated with systemic enchondromatosis has also been reported [14]. 

It is important to note that the presence of IDH1/2 mutations in glioma samples from enchondromatosis cases suggests a possible link between these two conditions [11]. The biomolecular studies performed in some reported cases of Ollier disease with brain glioma have shown a positive IDH1 mutation in a majority of cases. Corvino S et al. described 31 reported cases of Ollier disease with brain glioma, and biomolecular studies were performed only in 10 patients, where 8 patients showed positive IDH1-mutation [9]. Prokopchuk et al. described the non-skeletal neoplasms worldwide and found only 5 cases of astrocytoma associated with Maffucci syndrome [15]. According to previous studies, somatic mosaicism of IDH1/2 leads to the development of II-III grade gliomas in this disease [5,16,17]. This suggests that IDH1/2 mutations may be involved in the development of gliomas in patients with Ollier disease and other related disorders.

The presence of IDH1 mutations in astrocytomas is a common finding, and specifically, the c.395 G>A (IDH1 R132H) mutation is frequently observed in these tumors. Other genomic alterations, such as mutations in the ATRX, TP53, and CDKN2A/B genes, are also frequently observed in IDH-mutant astrocytomas [18,19]. Overall, the molecular profiling of IDH-mutant astrocytomas can provide important insights into the mechanisms underlying the development and progression of these tumors.

The hypothesis that somatic IDH mosaicism may be responsible for the development of gliomas in enchondromatosis patients is an interesting one. Bonnet et al. [5] suggested that the cartilaginous nature of these tumors may be related to the presence of IDH mutations, which are frequently observed in chondromas.

There have been a few reported cases of IDH1 R132C mutation in patients with Maffucci syndrome, and some of these cases were associated with glial tumors [5,15,16]. Additionally, IDH2 R172S and TP53 mutations have been identified in anaplastic astrocytomas in a patient with Maffucci syndrome [17]. These findings suggest that IDH mutations may play a role in the development of gliomas in patients with enchondromatosis and related disorders.

The case presented in this report is particularly interesting as it involves a grade IV astrocytoma with a c.395 G>A (IDH1 R132H) mutation. This mutation is one of the most common IDH mutations observed in gliomas [5,14,16,18], and its presence in this patient supports the hypothesis that IDH mutations may contribute to the development of gliomas in patients with enchondromatosis and related conditions. 

Identification of these mutations can aid in accurately classifying these disease entities and providing targeted treatment options for patients. However, it is noteworthy that although these tumors share similar appearances, the genes responsible for their development are diverse and have distinct functions, and there is no apparent close relationship between them in terms of their signaling pathways.

The absence of specific diagnostic and surveillance guidelines for individuals with Maffucci syndrome has resulted in the delayed diagnosis of certain malignancies, particularly gliomas. This underscores the importance of having tailored medical monitoring protocols to detect any potential tumors early. The identification of the common IDH1/2 mutation mechanism in gliomas and enchondromas offers a promising avenue for further research to understand the molecular mechanisms underlying these associations. This knowledge can help in the development of effective treatments for patiets with Maffucci syndrome who may be at an increased risk of developing these tumors.

Saiji et al. reported that next-generation sequencing (NGS) analysis enables the detection of IDH mutations in enchondromas, which suggests that most, if not all, central cartilaginous tumors could be treated using novel therapeutic approaches that target IDH mutations. Despite the limited sensitivity of IDH1 R132H antibody-based immunohistochemistry due to the diverse IDH1/2 variants found in enchondromas, it confirms the existence of intratumoral mosaicism along with NGS analysis [7].

While there is still much to be learned about these genetic mutations and their roles in tumor development and progression, targeted treatments that address these specific mutations hold significant potential for improving outcomes. 

The identification of common IDH mutations in both Maffucci syndrome and glial tumors, such as astrocytomas, presents a potential avenue for targeted therapy. By identifying the presence of IDH mutations in patients with Maffucci syndrome and associated glial tumors, physicians may be able to tailor treatment strategies that specifically target these mutations. This approach could potentially result in better treatment outcomes and reduced side effects compared to traditional treatments that are not specifically targeted to the genetic mutations present in these tumors.

However, it should be noted that further research is still needed to fully understand the effectiveness of targeted therapy for these patients. The rarity of the co-occurrence of Maffucci syndrome and glial tumors means that clinical trials are limited and it is difficult to draw definitive conclusions.

It is critical for individuals with Maffucci syndrome to receive regular medical monitoring and follow-up, including imaging studies, to enable early detection of any potential tumors. Furthermore, genetic counseling may be recommended to provide individuals with a comprehensive understanding of their condition and its associated risks.

## 4. Conclusions

Based on the presented case of grade IV astrocytoma with a c.395 G>A (IDH1 R132H) mutation, it is important to consider IDH1/2 mutations as a predictive marker for the development of gliomas. It is also necessary to establish appropriate monitoring protocols for patients with Maffucci syndrome to detect malignant tumors at an early stage.

Further studies can be conducted to better understand the relationship between IDH1/2 mutations and the development of gliomas, as well as to investigate potential treatments that can target these mutations. It is crucial to continue advancing our knowledge in this field to improve diagnosis and treatment for patients with gliomas and related conditions.

## Figures and Tables

**Figure 1 medicina-59-01056-f001:**
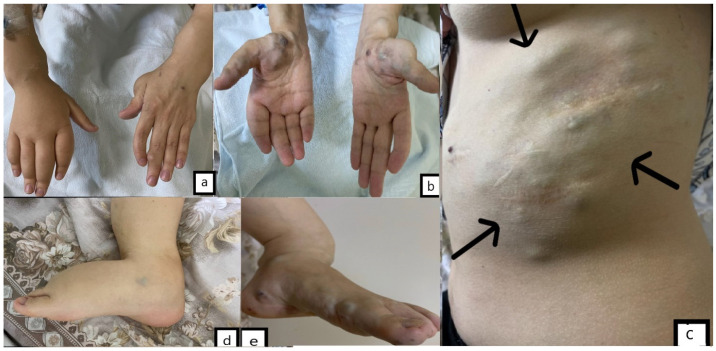
(**a**). Lymphedema of the right hand and cartilage lesion of the left thumb, phalanges deformity of the left hand. (**b**). Thumb and thenar hemangiomas (on both sides). (**c**). Hemangiomas of the abdominal wall and scars after hemangioma resections (arrows). (**d**). Lymphedema of the right foot. (**e**). Hemangiomas of the left foot.

**Figure 2 medicina-59-01056-f002:**
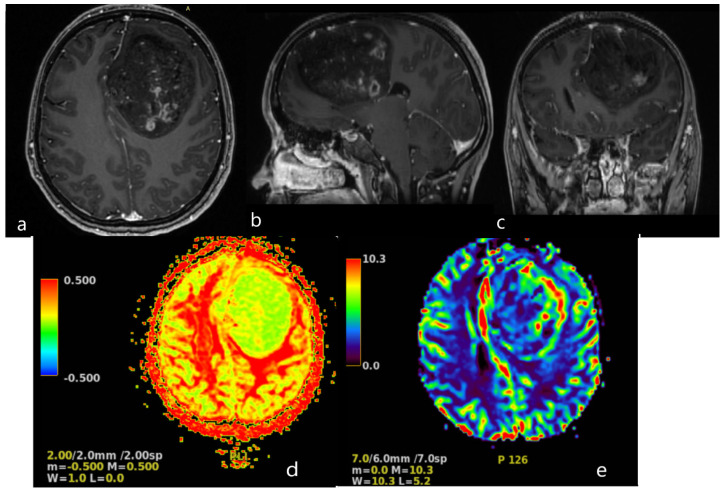
Brain MRI. Axial (**a**), sagittal (**b**) and frontal (**c**) contrast T1−weighted image demonstrates a large (83 × 58 × 62 mm) heterogeneously enhancing tumor in the left frontal lobe, results in a midine shift to the right side to 12 mm. Arterial Spin Labeled MRI Perfusion Imaging (**d**,**e**): elevated regional Cerebral blood perfusion (rCBF) and Cerebral blood volume (CBV) in lesion. Mean transit time (MTT) and Time to peak (TTP) is prolonged. MR spectroscopy (MRS): elevated choline/creatine peaks.

**Figure 3 medicina-59-01056-f003:**
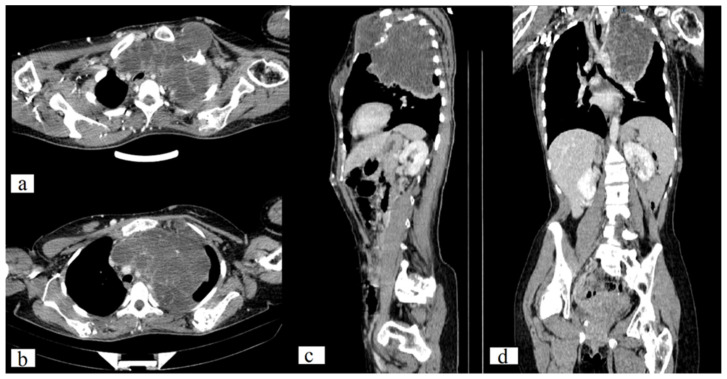
Axial (**a**,**b**), sagittal (**c**) and frontal (**d**) CT scan of the chest demonstrates a well-circumscribed, destructive lytic lesion of the left 1st rib (11.7 × 13.1 × 11.2 mm) and sternum (60 × 49 × 44 mm).

**Figure 4 medicina-59-01056-f004:**
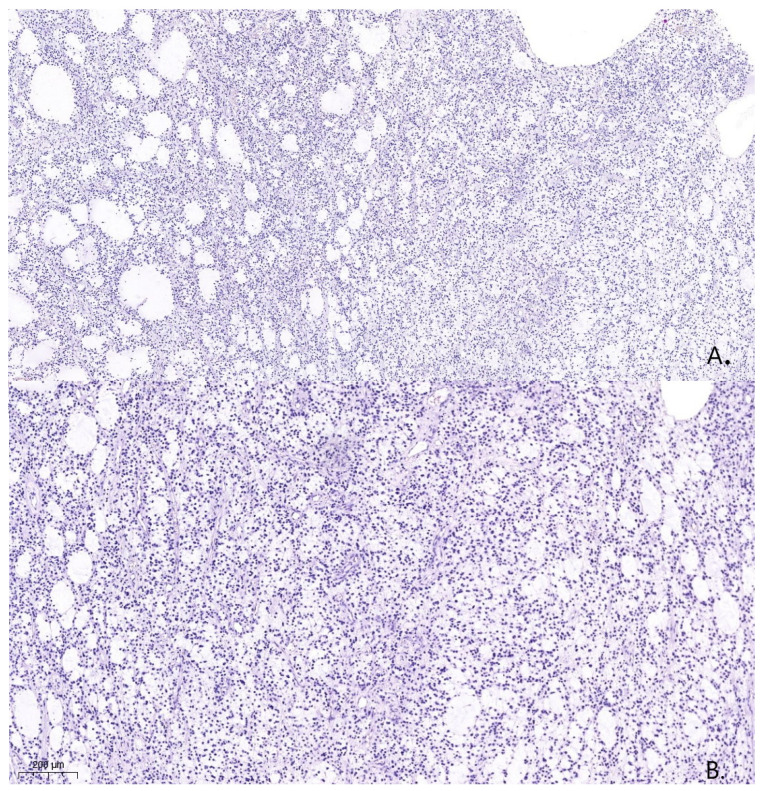
Histological specimen from the tumor resection demonstrates Astrocytoma, WHO grade 4, ICD-O code 9445/3. Hematoxylin and eosin. (**A**). (×100). (**B**). (×200).

## Data Availability

Not acceptable.

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
