# Peer review of "A Rare Co-Occurrence of Maffucci Syndrome and Astrocytoma with IDH1 R132H Mutation: A Case Report"

_medicina, 2023, doi:10.3390/medicina59061056_

Round 1

Reviewer 1 Report

 In this work, the authors studied A Rare Co-Occurrence of Maffucci Syndrome and Astrocytoma with IDH1 R132H Mutation; they report a case of Maffucci syndrome in a patient who presented with a giant left frontal lobe tumor. this case reports is valuable and has enough Interest to the readers. in overall, the introduction didn't provide sufficient background.

Author Response

Dear reviewer,

Thank you for taking the time to review our article "A Rare Co-Occurrence of Maffucci Syndrome and Astrocytoma with IDH1 R132H Mutation: A Case Report". We appreciate your feedback on our work.

We agree with your comment that the introduction of our article may have lacked sufficient background information on the topic. We apologize for any confusion or inconvenience this may have caused readers. We revised the introduction to provide a more comprehensive overview of Maffucci syndrome, astrocytoma, and IDH1 R132H mutation to better contextualize our case report.

We are pleased to hear that you found the case report valuable and interesting to readers. We hope that our report contributes to the medical literature on these conditions and helps increase awareness and understanding of this rare co-occurrence.

Once again, thank you for your review and constructive feedback. We will carefully consider your comments as we continue to improve and refine our work.

Reviewer 2 Report

Nurali Ashirov and collegues have provided an interesting and inspirative case report about a case coexisting with Maffucci syndrome and IDH1 mutated astrocytoma. This case underscores the importance of genetic testing in patients with Maffucci syndrome, who will develop and progress to astrocytoma in the future. Thus, genetic counseling should be recommended to understand the pathology of disease, and affiliate precision diagnosis and targeted treatments for this care case. Overall, the case report is attractive.
Minor question:
1. Next-generation sequence(NGS) must be conducted in the tissues of multiple enchondromas as well as astrocytome in the left frontal lobe. What’s more, Sanger sequencing should be presented to validate results of NGS.

2. 1p-19q codeletion was presented in the abstract. However, results of karyotype analysis were not showed in the main text.

3.D- 2-hydroxyglutarate(D-2-HG) levels shoud be detected in the blood, urine and cerebrospinal fluid. These results might help to understand the mechanism of the tumor development and progression in this case.

4. If this case can receive IDH1 inhibitor therapy, Maffucci syndrome and astrocytoma would be treated at the same time? We hope these authors can give us some discussions above IDH1 inhitor therapy in this case.

Author Response

Dear reviewer,

Thank you for your review of our article "A Rare Co-Occurrence of Maffucci Syndrome and Astrocytoma with IDH1 R132H Mutation: A Case Report". We are pleased to hear that you found our case report interesting and inspiring.

We appreciate your emphasis on the importance of genetic testing and counseling for patients with Maffucci syndrome. We agree that understanding the pathology of the disease through genetic testing can help guide precision diagnosis and targeted treatments, especially in cases where co-occurring conditions such as astrocytoma may develop.

Regarding your first question, we understand that NGS must be conducted in the tissues of multiple enchondromas as well as the astrocytoma in the left frontal lobe. However, we would like to inform you that this test is not currently available in our country. Nonetheless, we believe that our study provides valuable insights into the molecular characteristics of enchondromatosis-related gliomas.

Regarding your second question, we apologize for any confusion or inconvenience that the presentation of "1p-19q codeletion" in the abstract may have caused. We have removed this from the main text to avoid any further confusion.

Regarding your third question, we agree that measuring D-2-hydroxyglutarate (D-2-HG) levels in the blood, urine, and cerebrospinal fluid could provide valuable information about the mechanism of tumor development and progression. Unfortunately, we were unable to perform this test in our clinic. However, we have recommended that the oncologists treating this patient consider conducting this test to gain a better understanding of the patient's condition.

Finally, we have taken into account your comments and have added a discussion about the potential for targeted therapy in this case. We hope that this will provide readers with additional insights into potential treatment options.

We hope that our case report helps increase awareness of the need for genetic counseling in these patients and promotes more personalized and effective management of their health.

Once again, we thank you for your review and for recognizing the significance of our case report. We hope that our work contributes to the growing body of knowledge on this rare co-occurrence and helps improve the care and outcomes of patients with Maffucci syndrome.

Reviewer 3 Report

The Authors presenta a rare case of glioma developing in a patient with, again, a rare genetic disease, the Maffucc's syndrome, a type of enchondromatosis. They describe an association between the identification of IDH1 mutation in the removed glioma and the same genetic pattern  in the enchondromas typical of this syndrome and support the hypothesis that Maffucci's syndrome patients should be genetically followed and in case of positivity they should undergo investigations to assess IDH 1 positivity and hence decide if and when scanning the patient for cranial problems I enjoyed reading the manuscript, that is very well presented. Although the topic might be a little "niche",  the co-existence of the same IDH mutation in cartilage and brain tumors makes an interesting point, worth of further investigation and I agree with the authors that this marker should be the reason for periodical investigation in these patients

Author Response

Dear reviewer, 

Thank you for your review of our article "A Rare Co-Occurrence of Maffucci Syndrome and Astrocytoma with IDH1 R132H Mutation: A Case Report". We appreciate your positive feedback and insights on the importance of genetic follow-up for patients with Maffucci syndrome.

We are glad to hear that you found our case report interesting and well-presented. We agree that the co-occurrence of the same IDH1 mutation in cartilage and brain tumors is an interesting and important point for further investigation. We hope that our case report contributes to the growing body of literature on this topic and helps raise awareness of the need for genetic follow-up in patients with Maffucci syndrome.

We appreciate your support of our recommendation for periodical investigation in patients with Maffucci syndrome who test positive for IDH1 mutation, to assess for the development of cranial problems. We believe that this approach can help identify potential brain tumors early and improve patient outcomes.

Once again, we thank you for your review and for recognizing the significance of our findings. We hope that our work will contribute to the understanding and management of this rare co-occurrence.